# VALUE-BASED MEMBERSHIP INFERENCE ATTACK ON ACTOR-CRITIC REINFORCEMENT LEARNING

## ABSTRACT

In actor-critic reinforcement learning (RL), the so-called actor and critic, respectively, compute candidate policies and a value function that evaluates the candidate policies. Such RL algorithms may be vulnerable to membership inference attacks (MIAs), a privacy attack that infers the data membership, i.e., whether a specific data record belongs to the training dataset. We investigate the vulnerability of value function in actor-critic to MIAs. We develop *CriticAttack*, a new MIA that targets black-box RL agents by examining the correlation between the expected reward and the value function. We empirically show that *CriticAttack* can correctly infer approximately 90% of the training data membership, i.e., it achieves 90% attack accuracy. Such accuracy is far beyond the 50% random guessing accuracy, indicating a severe privacy vulnerability of the value function. To defend against *CriticAttack*, we design a method called *CriticDefense* that inserts uniform noise to the value function. *CriticDefense* can reduce the attack accuracy to 60% without significantly affecting the agent's performance.

## 1 INTRODUCTION

Membership inference attacks (MIAs) pose privacy vulnerabilities in reinforcement learning (RL) algorithms (Gomrokchi et al., 2020). Such attacks may make inferences about the training environments—whether a particular environment has been used in training—by observing the outcomes of an RL algorithm. For example, Pan et al. (2019); Wang et al. (2019); Chen et al. (2021) show that MIAs can infer users' vehicle routes or room layouts.

Most, if not all, existing methods for MIA suffer from high computational complexity or make unrealistic assumptions. For example, the methods in Pan et al. (2019) and Yang et al. (2021) rely on observing the learned policies. Both methods are computationally inefficient because they need to learn separate policies for each environment the attacker wants to infer. The methods in Gomrokchi et al. (2021; 2020) do not require learning additional policies for different environments, but assume that the attacker has full access to the RL algorithm, including the states, transitions, actions, and rewards on which the algorithm relies.

We propose a new black-box MIA called *CriticAttack* that alleviates the computational burden and relaxes the unrealistic assumptions made in the existing works. *CriticAttack* trains one set of policies for all environments, as opposed to training one set of policies per environment (e.g., Yang et al. (2021)). It makes inferences only based on the values generated by the value function and the expected rewards, in contrast to the states, transitions, actions, and rewards required by the existing work (e.g., Gomrokchi et al. (2021)).

We empirically show that *CriticAttack* can achieve 90% accuracy in inferring environments from the MiniGrid library (Chevalier-Boisvert et al., 2018). We perform the MIA on a state-of-the-art actor-critic RL algorithm (Schulman et al., 2017). The actor-critic algorithm trains two components: an actor and a critic. The actor generates policies that determine an RL agent's actions. The critic learns a value function that evaluates the policies by predicting the expected rewards, also known as rewards-to-go. The actor and the critic typically memorize their training environments (Haarnoja et al., 2018; Raichuk et al., 2021). Hence, we expect a high correlation between the values and the expected rewards from a training environment. On multiple RL tasks, *CriticAttack* achieves 90% attack accuracy, significantly higher than the 50% random guessing accuracy.

Such high attack accuracy is an indication of the severe privacy vulnerability of the value function.

We then turn our attention to defending against *CriticAttack*. We design a simple and efficient defense method called *CriticDefense* that concentrates on the value function. It inserts uniform noise to the value function to reduce the correlation between the values and the rewards-to-go. However, inserting noise introduces a trade-off between the attack accuracy and the agent's performance, e.g., measured by the cumulative reward that the agent obtains. *CriticDefense* can reduce the attack accuracy from 90% to 60% while degrading no more than 10% of the agent's performance.

Furthermore, we provide empirical evidence to show that the correlation between the values and the rewards-to-go is the primary source of privacy vulnerability. Due to the exploitation feature of RL, agents tend to choose the states experienced during training. The value function can accurately predict rewards-to-go on experienced states. Hence the correlation computed from a training environment is significantly higher than that from a test environment. The high correlation in the training environment leads to high attack accuracy.

The optimized value function plays a key role in transfer learning and the teacher-student framework. Many well-known transfer learning algorithms for actor-critic require the source agents to release their optimized value functions (Xu et al., 2020; Zhang & Whiteson, 2019; Takano et al., 2010). In the teacher-student framework, the student agents learn the optimal policies from the teacher's policies and value functions (Kurenkov et al., 2019). Therefore, it is essential to consider the privacy implications of the value function.

## 2 RELATED WORK

Pan et al. (2019) and Yang et al. (2021) develop MIA methods for deep RL that collect policies or actions for inference. While *CriticAttack* collects values from the value function and the cumulative reward for membership inference. Gomrokchi et al. (2021) and Gomrokchi et al. (2020) introduce two MIA methods to infer the roll-out trajectories in off-policy RL algorithms, which learn the optimal policy independently of the agent's actions. In contrast, *CriticAttack* works for on-policy RL algorithms, which optimize policies that determine what actions to take. From the defense perspective, several works (Garcelon et al., 2021; Lebensold et al., 2019b; Liao et al., 2021; Balle et al., 2016b; Chen et al., 2021) enforce differential privacy to the RL algorithm, which can protect against MIAs. Compared to the differential privacy mechanisms, we design *CriticDefense* for protecting the value function specifically. *CriticDefense* provides robust protection against attacks on the value function; however, it has limited ability to protect other components in the algorithm and does not achieve differential privacy.

## 3 PRELIMINARY

**Reinforcement Learning (RL)** is an area of machine learning where we train an agent or a set of agents by interacting with a set of environments. The agent observes a state from the environment, then takes action based on its policy $\pi$, and receives a reward from the environment that evaluates this action.

We formally define the *environment* as a Markov decision process (MDP) $E = \{S, A, P, I, R\}$, where $S$ and $A$ are the sets of states and actions, $P : S \times A \mapsto S$ is the state transition function, $I : S \mapsto [0, 1]$ is the initial distribution of the states, and $R : S \mapsto \mathbb{R}$ is the reward function. We consider a set of environments as the training dataset of the agent that may face privacy threats.

**Actor-Critic (Konda & Tsitsiklis, 1999)** is one of the state-of-the-art RL algorithms that trains two components: actor and critic. The actor with parameters $\theta$ takes the current state representation and all possible actions as input and then generates a policy $\pi_\theta$. The critic $V^\pi(s)$ with parameters $\phi$ learns a value function, which takes the current state observation as input and outputs a *value* that evaluates the actions leading to the current state. We present the details of the actor-critic algorithm in the Appendix.

In the training stage, we run the agent in the set of environments to collect a set of *trajectories*. Each trajectory consists of a sequence of tuples (state $s_t$, action $a_t$, reward $r_t$, new state $s_{t+1}$) with respect to timestamp $t = 1, ..., T$. We then estimate the advantage $A_t^\pi$ to compute the parameters' gradient in both the actor and the critic. Most actor-critic algorithms use one of the three advantage estimation methods in Equation 1.

The value function evaluates the current state and past actions leading to the current state by estimating the *reward-to-go*. Reward-to-go is the expected cumulative reward the agent can get if starting from the current state: $\tilde{r}_t = \sum_{k=t}^{T} r_k$. Note that $r_k : k > t$ is an expected reward the agent will likely to get at timestamp $k$. In the training stage, we compute the reward-to-go at timestamp $t$ by giving the values and the advantage estimation method $\mathcal{E}$:

$$\text{TD advantage: } \tilde{r}_t = A_t^\pi + V_\phi^\pi(s_t), \quad A_t^\pi = r_t + \gamma V_\phi^\pi(s_{t+1}) - V_\phi^\pi(s_t).$$

$$\text{N-step advantage: } \tilde{r}_{t,N} = A_{t,N}^\pi + V_\phi^\pi(s_t), \quad A_{t,N}^\pi = \sum_{k=0}^{N-1} \gamma^k r_{t+k+1} + \gamma^N V_\phi^\pi(s_{t+N+1}) - V_\phi^\pi(s_t). \tag{1}$$

$$\text{Generalized advantage: } \tilde{r}_{t,\lambda} = A_{t,\lambda}^\pi + V_\phi^\pi(s_t), \quad A_{t,\lambda}^\pi = \sum_{k=t}^{T} (\gamma\lambda)^{k-t} A_k^\pi.$$

We train the critic to estimate the reward-to-go at a given state $s_t$, and update the parameters $\phi$ accordingly to minimize the value loss:

$$\mathcal{L}(\phi) = \sum_t ||V_\phi^\pi(s_t) - \tilde{r}_t||^2. \tag{2}$$

**Membership Inference Attack (MIA)** is one of the well-known privacy attacks that can be applied to machine learning models to infer whether a selected data record belongs to the training dataset of the given model. The *shadow model* framework (Shokri et al., 2017) is the standard approach to MIAs on machine learning models, where the shadow models mimic the behavior of the *target model*. Since the training datasets of the shadow models are known, the attacker can learn to infer whether a data record is used in training the shadow model. We then apply the trained attacker to infer the target model. We denote the percentage of the correctly inferred data records as the *attack accuracy*.

## 4 ATTACK METHOD

We design an *environment-based* MIA on actor-critic algorithms named *CriticAttack*. In environment-based (user-based) MIA, the attacker infers about an environment, as opposed to trajectory-based (sample-based) MIA, where the attacker infers about a single trajectory. *CriticAttack* determines whether the agent has been trained under a particular environment based on the observation of the values and rewards-to-go.

### 4.1 ASSUMPTION

A *target agent* is the RL agent trained by a set of private environments that the attacker wants to infer. In this work, we perform *CriticAttack* on the target agent whose policies and value functions are composed of neural networks and optimized by the actor-critic algorithm.

The information of the target agent includes the well-trained parameters of the actor-network and the critic-network, the specifications of the two networks such as the number of layers and the activation function, the training algorithm with hyper-parameters, loss functions, the gradient history, and feedback from the environments.

The attacker typically does not have full access to the information of the target agent. Based on the attacker's access, we can categorize MIAs into two groups: black-box attack and white-box attack (Hu et al., 2021). The black-box attacker only has access to the inputs and outputs of the neural networks, the actor-network and the critic network in this case. In contrast, the white-box attacker has full access to the parameters of the neural networks, loss functions, and gradients. However, several black-box MIAs (Shokri et al., 2017; Sablayrolles et al., 2019) can also access the network specifications, training algorithm, and hyper-parameters.

To distinguish black-box and white-box attacks, we consider the access to the networks' parameters and gradient history as the borderlines between the two types of attacks. The attacker is a black-box attacker if it has access to neither networks' parameters nor gradient history.

In this work, we assume the attacker only has access to the inputs and outputs of the actor and critic networks, the training algorithm (including the advantage estimation method), hyperparameters, and rewards from the environments. Therefore, *CriticAttack* falls into the category of *black-box attack*.

## 4.2 CRITICATTACK

*CriticAttack* follows the *shadow model* framework (Shokri et al., 2017). Since we do not have access to the target agent's training environments, we train a set of shadow agents with known environments to mimic the behavior of the target agent. The attacker learns how the shadow agents behave differently in visited and new environments.

We assume there is a public universal data distribution that all the environments, regardless of whether they are used during training, are drawn from this distribution. So, the attacker can obtain similar datasets to train the shadow agents. Once the attacker learns to differentiate whether an environment has been used in training the shadow agents, we can apply it to the target agent.

Training the attacker takes the following three steps:

***First***, we obtain a set of environments from the data source and evenly partition the environments into two groups: training environments and validation environments. We train each shadow agent using the training environments until its performance is less than 5% different from the target agent. We measure the performance by the average rewards in the validation environments. We then repeat this step to construct multiple shadow agents.

***Second***, we run each shadow agent on its training and validation environments to collect trajectories. For each environment $E$, we collect a corresponding trajectory set $S_E$ that contains *n critic trajectories*, which is defined in Definition 4.1.

**Definition 4.1.** A *critic trajectory* $T^{v,\tilde{r}}$ consists a sequence of (value, reward-to-go) tuples:

$$T^{v,\tilde{r}} = \{(V_\phi^\pi(s_t), \tilde{r}_t) : t = 0, ..., T\},$$

where $T$ is the trajectory length. The critic trajectory can break up into a *value trajectory* $T^v$ and a *reward-to-go trajectory* $T^{\tilde{r}}$:

$$T^v = \{V_\phi^\pi(s_t) : t = 0, ..., T\}, \quad T^{\tilde{r}} = \{\tilde{r}_t : t = 0, ..., T\}.$$

Note that we need to compute the rewards-to-go given the rewards and the value trajectory. We trace the rewards $r = \{r_1, ..., r_T\}$ and the value trajectory $T^v$ from $T$ to 0 to compute the reward-to-go trajectory $T^{\tilde{r}}$ and get the critic trajectory $T^{v,\tilde{r}}$ using Algorithm 1. After obtaining the critic trajectories, we label each critic trajectory set $S_E$ as 'in' if $E$ belongs to the training environments and 'out' otherwise. By repeating the second step, we get a critic trajectory set and its label for every environment in the training and validation dataset. These critic trajectories and labels form a supervised learning dataset for the attacker.

***Third***, we train a binary classifier that takes a set of critic trajectories $S_E$ as input and determines the corresponding environment $E$ is 'in' or 'out' of the training environments. We design two architectures for the binary classifier: the logistic regression classifier and the deep neural network classifier.

*Logistic Regression on Correlation Score (LR)* focuses on the correlation between values and rewards-to-go. Suppose we have collected $N$ sets of critic trajectories from the shadow agents, where each set contains $n$ trajectories. For each environment $E$, we extract the value trajectories $T_i^v$ and the reward-to-go trajectories $T_i^{\tilde{r}}$ from the trajectory set $S_E$, where $i = 1, ..., n$ and $n$ is number of trajectories in $S_E$. Then, we compute the average correlation between the value trajectories and reward-to-go trajectories following Equation 3:

---

**Algorithm 1:** REWARD-TO-GO ESTIMATION

---

**Input:** value trajectory $T^v$, rewards $r$, discount factor $\gamma$, estimation method $\mathcal{E}$,
   hyper-parameters $N, \lambda$
**Output:** critic trajectory $T^{v,\tilde{r}}$
$A^\pi_{T+1,N,\lambda}, V^\pi_\phi(s_{T+1}), r_{t>T} = 0, 0, r_T$
**for** $t = T$ *to* $0$ **do**
 **if** $\mathcal{E}$ *is TD advantage* **then**
  $A^\pi_{t,N,\lambda} = r_t + \gamma V^\pi_\phi(s_{t+1}) - V^\pi_\phi(s_t)$ ;     /* refer to Equation 1*/
 **end**
 **if** $\mathcal{E}$ *is N-step advantage* **then**
  $A_t = r_{t+1} - V^\pi_\phi(s_t) + \gamma V^\pi_\phi(s_{t+1})$ ;    /* refer to Equation 1*/
  $A_{t+N} = r_{t+N+1} - V^\pi_\phi(s_{t+N+1}) + \gamma V^\pi_\phi(s_{t+N+2})$
  $A^\pi_{t,N,\lambda} = \gamma A^\pi_{t+1,N,\lambda} + A_t - \gamma^N A_{t+N}$ ;   /* Proof:  see Appendix*/
 **end**
 **if** $\mathcal{E}$ *is Generalized advantage* **then**
  $A^\pi_{t,N,\lambda} = r_t + \gamma V^\pi_\phi(s_{t+1}) - V^\pi_\phi(s_t) + \gamma \lambda A^\pi_{t+1,N,\lambda}$ ;  /* refer to Equation 1*/
 **end**
 $\tilde{r}_t = A^\pi_{T+1,N,\lambda} + V^\pi_\phi(s_t)$
**end**
$T^{v,\tilde{r}} = \{(v_t, \tilde{r}_t) : t = 0, ..., T\}$

---

$$\rho_E = \frac{1}{n}\sum_{i=1}^{n}\frac{cov(T_i^v, T_i^{\tilde{r}})}{\sigma_{T_i^v}\sigma_{T_i^{\tilde{r}}}} = \frac{1}{n}\sum_{i=1}^{n}\frac{\mathbb{E}[(T_i^v - \mu_{T_i^v})(T_i^{\tilde{r}} - \mu_{T_i^{\tilde{r}}})]}{\sigma_{T_i^v}\sigma_{T_i^{\tilde{r}}}}. \tag{3}$$

In the training stage, we compute a correlation score $\rho_E$ for each environment $E$, form a (correlation, label) tuple, and mark the label 1 to represent 'in' and 0 to represent 'out.' We then fit the logistic regression classifier with all the (correlation, label) tuples.

In the inference stage, we compute the average correlation score $\rho_{E_{val}}$ for the target agent on a given environment $E_{val}$ and use the logistic regression classifier to predict if $E_{val}$ belongs to the training dataset of the target agent.

*Deep Neural Network (DNN)* takes the concatenation $\oplus$ of the value trajectory and the corresponding reward-to-go trajectory as input and performs binary classification:

$$NN_\omega(T_i^v \oplus T_i^{\tilde{r}}) = \begin{cases} 1, & \text{as 'in',} \\ 0, & \text{as 'out'.} \end{cases}$$

In the training stage, we assign a label 'in' or 'out' to each critic trajectory depending on whether it is collected from a training environment. We then train the neural network with these labeled trajectories.

In the inference stage, we run the target agent on a given environment $E$ to obtain $n$ trajectories. We apply the trained neural network to predict each trajectory and take the majority vote as the prediction for the given environment.

## 5 DEFENSE METHOD

In practice, the best protection is to conceal the value function from users. However, RL agents that allow users to fine-tune or allow to be the teacher in transfer learning must release their value functions. In such scenarios, we introduce a defense method named *CriticDefense* specifically against *CriticAttack*. We consider the correlation between the values and the rewards-to-go as one of the primary factors impacting the attacker's decision. Therefore, we develop *CriticDefense* to modify the correlations and examine if it can effectively reduce the attack accuracy.

## 5.1 CRITICDEFENSE

*CriticDefense* inserts uniform noise into a certain percentage of the values during training. Practically, we replace the value function $V_\phi^\pi$ in the actor-critic algorithm with the following:

$$\tilde{V}_\phi^\pi(s_t) = (V_\phi^\pi(s_t) + \mathbb{1}_{[0,\mathscr{R})}(u) \cdot u \cdot \mathscr{R})\%1, \tag{4}$$

where $\mathscr{R}$ is a hyper-parameter to define the noise percentage, $u$ is sampled from a standard uniform distribution: $u \sim U(0,1)$, % is the modulo operation. *CriticDefense* adds negligible noise to a small proportion of the values as $\mathscr{R}$ approaches 0 and adds large noise to almost all the values if $\mathscr{R}$ approaches 1.

Adding uniform noise to the values indirectly adds noise to the rewards-to-go since we compute rewards-to-go based on the values according to Algorithm 1. *CriticAttack* makes inference only based on the values and rewards-to-go; adding noise to both components is the most straightforward approach to protect against such attack.

In RL algorithms, values are strictly between 0 and 1, so we use a modulo to guarantee this. Compared to clipping the noisy values, modulo one ensures the noisy values do not exceed the upper limit and allows a more significant change, which may further break the correlation between the values and rewards-to-go. For instance, the original value of 0.95 with a noise of 0.1 will result in an invalid value of 1.05. Value clipping clips the noisy value to 1 and makes it valid. In contrast, modulo results in a new value of 0.05. The modulo will be triggered more constantly as $\mathscr{R}$ approaches 1.

## 5.2 COMPATIBILITY

*CriticDefense* is designed to protect the value function against *CriticAttack*, so it has limited efficacy in protecting the policies and mitigating overfitting. We integrate two other methods with *CriticDefense* to strengthen the protection of other components of the actor-critic algorithm, such as policies. *CriticDefense* is compatible with the two methods below, which means they do not interfere with each other and introduce extra performance loss.

**Regularization** is a prominent approach to prevent overfitting by lowering the complexity of the neural networks during training (Kukačka et al., 2017). Many works have demonstrated that regularization reduces MIA accuracy by mitigating overfitting (Ying et al., 2020; Nasr et al., 2018; Kaya et al., 2020). We consider applying L2 regularization in the actor-critic algorithm: add a regularization loss with a regularization rate $a$ to the original value loss $\mathscr{L}(\phi)$.

**Value Clipping** is another approach to prevent the value function from being over-adapted to the newly added training environment and losing the information from previous environments (Schulman et al., 2017). Existing work has shown its effectiveness in protecting against MIA in RL policies (Yang et al., 2021). It clips the norm of the value loss to $\epsilon_{clip}$ to restrict the step size of updating the parameters.

## 6 EMPIRICAL ANALYSIS

In this section, we perform two sets of MIA experiments. In the first set of experiments, we provide empirical evidence to show that *CriticAttack* can correctly infer above 90% of the training environments. In the second set of experiments, we demonstrate the vulnerability factor of CriticAttack and show that *CriticDefense* can reduce the attack accuracy to 60% while maintaining the agent's performance.

In both the attack and defense section, we clip the values to $\epsilon_{clip} = 0.2$ and enforce L2-regularization with $a = 0.01$ to mitigate overfitting while training the agents. We apply the two methods in both sections to show that: 1) *CriticAttack* works well even if the mitigation methods are applied, and 2) the attack accuracy drops due to *CriticDefense* rather than the two mitigation methods.

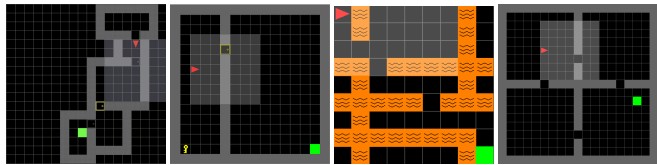

Figure 1: RL tasks from left to right: Multi-Rooms, Door-Key, Lava-Crossing, Four-Rooms.

**Environment Setup**  In all the experiments, we use the MiniGrid toolkit (Chevalier-Boisvert et al., 2018) as the underlying testbed. We choose four tasks listed in Figure 1, wherein the agent learns to reach a target destination without bumping into obstacles. Once the agent reaches the destination or a fixed number of steps, the environment resets to a new map while the task remains the same. We control the map's layout by fixing random seeds due to the one-to-one correspondence between seeds and maps. We use the maps to simulate room/warehouse layouts in real-world settings and reveal their privacy. Assuming that each new map represents a floor map of private property, we perform MIA to infer whether the agent has visited a given floor map in its training, which violates the privacy of the private properties.

**Experiment Setup**  We have presented three different advantage estimation methods in Equation 1, so we perform three subsets of experiments for the three estimation methods separately. We use the Proximal Policy Optimization (PPO) algorithm (Schulman et al., 2017) for deploying the three advantage estimations: TD advantage, N-step advantage (N-Step), and Generalized advantage (GAE). We apply the PPO algorithm on 40 unique maps to train each target agent (we also show the MIA results on targets with more training maps in the Appendix). We implement the PPO algorithm using the RL-Starter-Files library (Willems, 2018) with default hyper-parameters unless specified below.

### 6.1 ATTACK

Following *CriticAttack* in the Methodology section, we train five shadow agents for each target agent. Due to the assumption that the attacker does not have access to the training data size, we use 20 distinct maps to train each shadow agent until it converges to the same reward as the target agent.

Then, we apply each shadow agent to 40 distinct maps- in which 20 maps are used to train this shadow agent- to collect 25 critic trajectories from each map. So, we can collect 200 trajectory sets with 5,000 critic trajectories from 5 shadow agents and use them for training the attacker. Once we finish training the attacker, we apply it to the target agent to infer the training maps of the target agent.

We have proposed two architectures for the attacker: LR and DNN. We present the MIA results of the two architectures in Table 1 and show some visualized examples in Figure 2. The LR attacker can achieve approximately 90% accuracies by only finding a correlation threshold. The DNN attacker can get close to 95% accuracies; however, it is computationally inefficient compared to the LR. In summary, both attackers demonstrate the severe vulnerability of the value function by showing such high attack accuracies.

| Estimation(Attacker) | Multi-Rooms | Door-Key | Lava-Crossing | Four-Rooms |
|---|---|---|---|---|
| TD Advantage (LR) | $90.5 \pm 1.6$ | $89.2 \pm 1.3$ | $91.8 \pm 0.9$ | $93.1 \pm 0.8$ |
| TD Advantage (DNN) | $95.2 \pm 1.1$ | $92.9 \pm 1.2$ | $93.8 \pm 1.2$ | $96.2 \pm 0.5$ |
| N-Step (LR) | $89.7 \pm 1.4$ | $88.7 \pm 1.6$ | $90.6 \pm 1.2$ | $91.5 \pm 1.1$ |
| N-Step (DNN) | $94.2 \pm 0.8$ | $91.4 \pm 1.4$ | $93.2 \pm 1.3$ | $94.8 \pm 0.6$ |
| GAE (LR) | $89.7 \pm 1.4$ | $89.7 \pm 1.6$ | $90.2 \pm 1.3$ | $92.5 \pm 1.0$ |
| GAE (DNN) | $95.7 \pm 1.3$ | $94.1 \pm 1.3$ | $94.2 \pm 0.9$ | $97.1 \pm 0.6$ |

Table 1: MIA accuracies (mean $\pm$ standard deviation) across five repetitions.

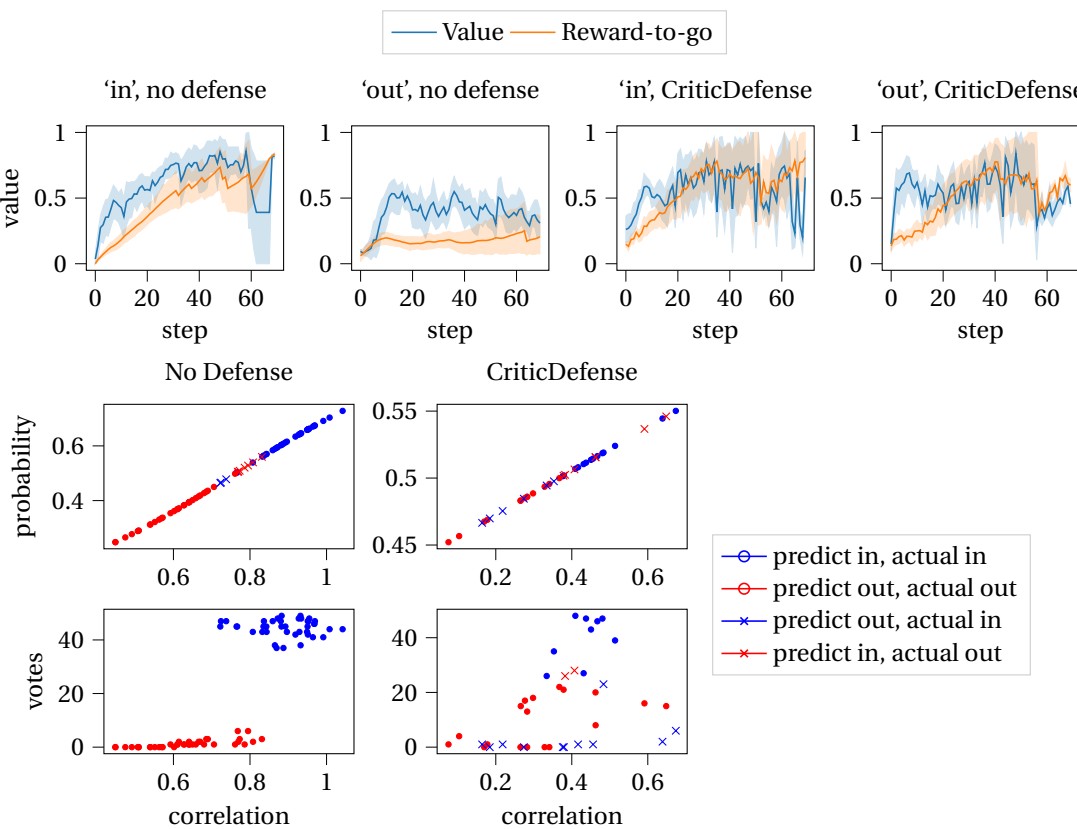

Figure 2: MIA on a PPO algorithm with GAE trained for the Multi-Rooms task. The figures in the first row show the value and rewards-to-go trajectories. The second and third rows show the MIA results using LR and DNN, respectively.

## 6.2 DEFENSE

We now investigate the effects of *CriticDefense* and compare it with the well-known differential privacy mechanism DP-SGD. We assume the attacker knows the defense methods applied to the target model, so we also deploy the defense methods with the same parameters while training the shadow models. Note that we are not trying to indicate that the *CriticDefense* outperforms DP-SGD. *CriticDefense* is more suitable for protecting the value function against *CriticAttack*, but DP-SGD is potentially more effective in protecting the policies.

**Privacy-Performance Trade-off** Figure 3 shows the defense results on GAE on the Multi-Rooms task, while the MIA settings are identical to Section 6.1. We observe that both of the defense methods can reduce attack accuracies. *CriticDefense* can reduce the attack accuracy to approximately 60% with less than 10% performance loss, measured by rewards. We compute the performance loss using the cumulative reward of the unprotected agent $r_T$ and the cumulative reward of the protected agent $r'_T$: $\mathcal{L}_{perf} = \frac{r_T - r'_T}{r_T}$. *CriticDefense* can reduce the attack accuracies to around 60% with approximately 10% of performance losses in the other tasks, regardless of the estimation methods. Figure 2 compares attack results before and after applying *CriticDefense* and shows how *CriticDefense* reduces the attack accuracy. We also present the numerical results in Table 4 in the Appendix.

In contrast, the DP-SGD algorithm can reduce the MIA accuracy to approximately 50% with over 60% performance loss. Additionally, the actor-critic algorithm with DP-SGD requires a significantly larger number of steps to convergence than *CriticDefense*. It means the computation of training a protected agent is ten times more expensive than training an unprotected one.

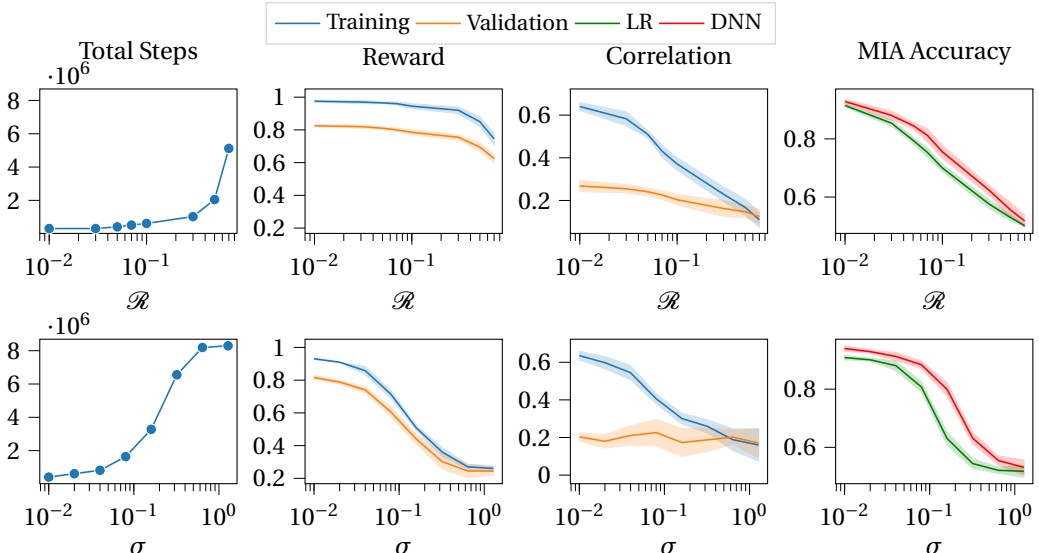

Figure 3: Protections against MIA. The first row shows the results of *CriticDefense*, and the second row shows the results of DP-SGD. The columns from left to right present the required number of steps to convergence, final rewards upon convergence, the average correlation between values and rewards-to-go, and attack accuracies. In DP-SGD, we use the noise variance $\sigma$ as the x-axis. We define the differential privacy budget $\epsilon, \delta$, and plot $\sigma$ vs. $\epsilon$ in the Appendix.

### 6.3 VULNERABILITY FACTOR

Due to the exploitation feature of RL, when we place an agent in its training environment, the agent tends to take the 'best' trajectory that it has experienced during training. Therefore, the value function can accurately predict the rewards-to-go, causing the correlations from training environments to be significantly higher than correlations from validation environments. Hence we can achieve above 85% attack accuracy simply by using a logistic regression classifier to find the threshold between the training and validation correlations.

*CriticDefense* significantly reduces the training correlation and shrinks the gap between the training correlation and validation correlation. We can observe that the attack accuracies are decreasing as the training and validation correlations approach each other. Therefore, we conclude that the correlation between values and rewards-to-go is the primary source of privacy vulnerability to *CriticAttack*.

Since *CriticDefense* can reduce the correlation by only adding a small amount of noise (e.g., $\mathscr{R} = 0.3$), it is sufficient to protect the value function against *CriticAttack* while maintaining the agent's performance. We also present specific examples to support our claims in the Appendix.

## 7 CONCLUSION

In this work, we introduce an effective and efficient black-box membership inference attack named *CriticAttack* that concentrates on the value function of the actor-critic algorithm. We empirically demonstrate the high vulnerability of the value function of the actor-critic algorithm to MIAs by showing approximately 90% attack accuracies. Therefore, RL services should provide users with the least possible access to the value function. We then design a corresponding defense method called CriticDefense, which can significantly reduce the attack accuracies of *CriticAttack* without hurting the target agent's performance. A limitation of the current work is that *CriticAttack* only works for actor-critic algorithms. We can generalize this MIA to other reinforcement learning algorithms consisting of value functions as a future direction.

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

# A APPENDIX

## A.1 ACTOR-CRITIC ALGORITHM

---

**Algorithm 2:** ACTOR-CRITIC

---

**Input:** policy $\pi_\theta$, value function $V_\phi^\pi$, environment $E = \{S, A, P, I, R\}$, discount factor $\gamma$,
      learning rate $\alpha$, episode $\tau$
**Output:** optimized policy $\pi_\theta$, optimized value function $V_\phi^\pi$

**for** $k = 1$ *to* $\tau$ **do**
    Initialize first state $s_0$, timestamp $t = 0$
    **while** $s_t$ *is not terminal* **do**
        $a_i \sim \pi_\theta(a_t|s_t), s_{t+1} = P(s_t, a_i), r_t = R(s_t)$;   /* take an action $a_i$, observe the next state $s_{t+1}$ and the reward $r_t$*/
        Compute $V_\phi^\pi(s_t)$
        compute advantage $A_t^\pi$ and reward-to-go $\tilde{r}_t$;     /* varied by estimation methods, stated in Equation 1*/
        $\theta = \theta + \alpha \nabla_\theta \log \pi_\theta(a|s) A_t^\pi$;        /* update parameters in actor*/
        $\phi = \phi + \alpha \nabla_\phi (V_\phi^\pi(s_t) - \tilde{r}_t)^2$;       /* update parameters in critic*/
        $t = t + 1$
    **end**
**end**

---

## A.2 CRITICATTACK PIPELINE

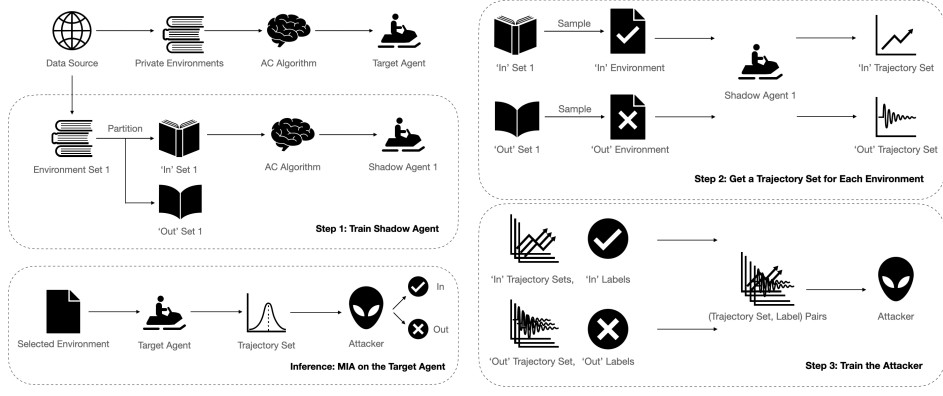

Figure 4: CriticAttack Pipeline.

## A.3 PROOF OF THE N-STEP ADVANTAGE IN ALGORITHM 2

**Theorem A.1.** In the N-Step Advantage, given the $\{t+1\}^{th}$ advantage $A_{t+1,N}^\pi$, then the $t^{th}$ advantage is

$$A_{t,N}^\pi = \gamma A_{t+1,N}^\pi + \left( r_{t+1} - V_\phi^\pi(s_t) + \gamma V_\phi^\pi(s_{t+1}) \right)$$
$$- \gamma^N \left( r_{t+N+1} - V_\phi^\pi(s_{t+N+1}) + \gamma V_\phi^\pi(s_{t+N+2}) \right). \tag{5}$$

*Proof.* We have shown the N-Step Advantage in the Background section:

$$A_{t,N}^\pi = \sum_{k=0}^{N-1} \gamma^k r_{t+k+1} + \gamma^N V_\phi^\pi(s_{t+N+1}) - V_\phi^\pi(s_t)$$
$$= r_{t+1} + \gamma r_{t+2} + \ldots + \gamma^{N-1} r_{t+N}$$
$$+ \gamma^N V_\phi^\pi(s_{t+N+1}) - V_\phi^\pi(s_t). \tag{6}$$

The $\{t+1\}^{th}$ advantage is

$$
\begin{aligned}
A_{t+1,N}^{\pi} &= \sum_{k=0}^{N-1} \gamma^k r_{t+k+2} + \gamma^N V_{\phi}^{\pi}(s_{t+N+2}) - V_{\phi}^{\pi}(s_{t+1}) \\
&= r_{t+2} + \gamma r_{t+3} + \dots + \gamma^{N-1} r_{t+N+1} \\
&\quad + \gamma^N V_{\phi}^{\pi}(s_{t+N+2}) - V_{\phi}^{\pi}(s_{t+1}).
\end{aligned}
\tag{7}
$$

Then, we can compute

$$
\begin{aligned}
\gamma A_{t+1,N}^{\pi} &= \gamma r_{t+2} + \gamma^2 r_{t+3} + \dots + \gamma^N r_{t+N+1} \\
&\quad + \gamma^{N+1} V_{\phi}^{\pi}(s_{t+N+2}) - \gamma V_{\phi}^{\pi}(s_{t+1}),
\end{aligned}
\tag{8}
$$

hence

$$
\begin{aligned}
A_{t,N}^{\pi} - \gamma A_{t+1,N}^{\pi} &= r_{t+1} - \gamma^N r_{t+N+1} + \gamma^N V_{\phi}^{\pi}(s_{t+N+1}) \\
&\quad - V_{\phi}^{\pi}(s_t) - \gamma^{N+1} V_{\phi}^{\pi}(s_{t+N+2}) + \gamma V_{\phi}^{\pi}(s_{t+1}),
\end{aligned}
\tag{9}
$$

$$
\begin{aligned}
A_{t,N}^{\pi} &= \gamma A_{t+1,N}^{\pi} + \left( r_{t+1} - V_{\phi}^{\pi}(s_t) + \gamma V_{\phi}^{\pi}(s_{t+1}) \right) \\
&\quad - \gamma^N \left( r_{t+N+1} - V_{\phi}^{\pi}(s_{t+N+1}) + \gamma V_{\phi}^{\pi}(s_{t+N+2}) \right).
\end{aligned}
\tag{10}
$$

$\square$

## A.4 COMPARISON BETWEEN EXISTING MIAS

| Attacker | Multi-Rooms | Door-Key | Lava-Crossing | Four-Rooms |
|---|---|---|---|---|
| Pan et al. (2019) | $82.3 \pm 1.9$ | $79.5 \pm 2.3$ | $78.4 \pm 2.5$ | $87.2 \pm 1.6$ |
| Yang et al. (2021) | $93.9 \pm 1.2$ | $93.5 \pm 0.7$ | $91.6 \pm 1.5$ | $96.1 \pm 0.4$ |
| CriticAttack (LR) | $89.7 \pm 1.4$ | $89.7 \pm 1.6$ | $90.2 \pm 1.3$ | $92.5 \pm 1.0$ |
| CriticAttack (DNN) | $95.7 \pm 1.3$ | $94.1 \pm 1.3$ | $94.2 \pm 0.9$ | $97.1 \pm 0.6$ |

Table 2: MIA accuracies (mean ± standard deviation) across five repetitions. We use GAE for all the results.

## A.5 MORE EXAMPLES OF CRITICATTACK

We present more examples of CriticAttack on all three advantage estimation methods in Figure 5. We also show the value trajectories and reward-to-go trajectories collected from a selected environment which is 'in' agent 1's training dataset but 'out' of agent 2's training dataset. We can observe the value and reward-to-go trajectories generated by agent 1 are highly correlated compared to agent 2.

We also perform *CriticAttack* on target agents whose training data sizes are varied. We attack the target agents trained using 10, 20,..., and 100 environments and observe how the training data size affects the attack accuracy. We train the shadow models using identical numbers of training environments as the target model. We present the results in Figure 6.

We can observe a negative correlation between the training data size and the attack accuracy. Increasing the training data size will improve the RL agent's generalization power, reducing the attack accuracy. However, the impact of training data size to attack accuracy is insignificant. We can only reduce the attack accuracy by approximately 5% by doubling the training data size. Simultaneously, the training time significantly increases while we have more training data.

## A.6 DIFFERENTIAL PRIVACY

Differential Privacy is a technique for describing the overall patterns of the dataset while concealing the individual information. It guarantees that two *adjacent* sets of objects have an indistinguishable impact on the overall outcomes by adding a proper amount of statistical noise to the raw data.

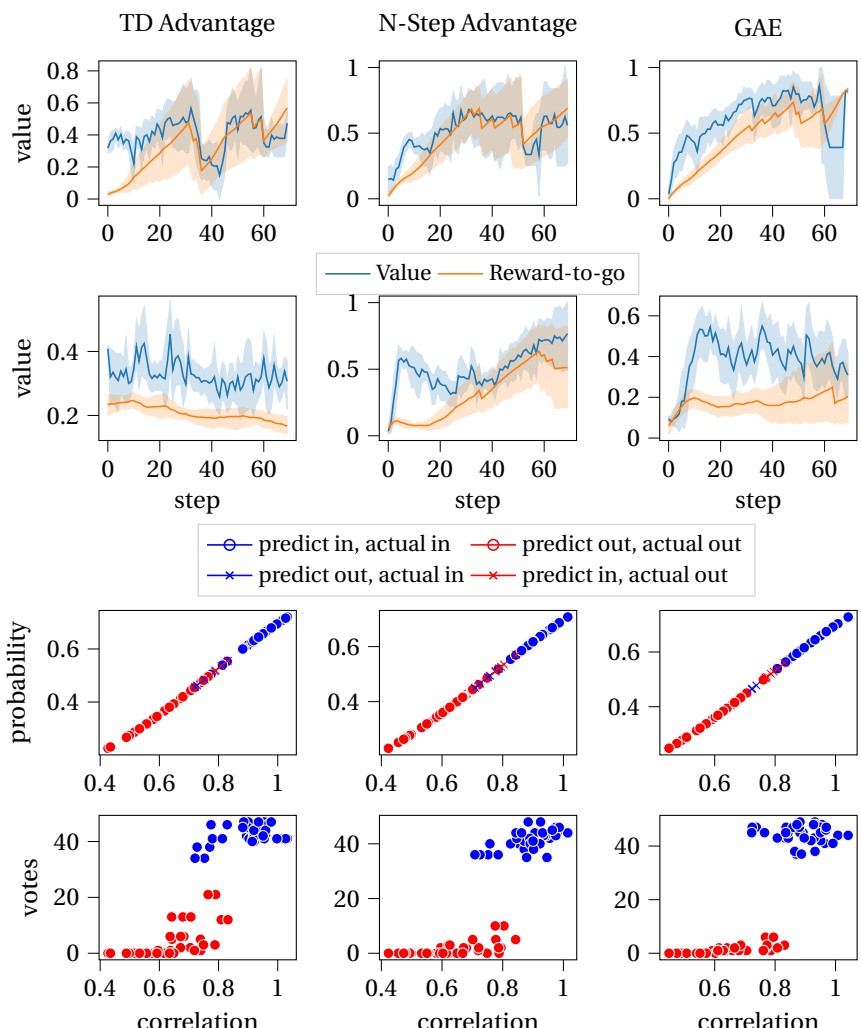

Figure 5: The first and second rows show the value and reward-to-go trajectories collected from a selected environment $E_0$, generated by two separate agents: agent 1 and agent 2. Note that $E_0$ is in the training dataset of agent 1 but out of the training dataset of agent 2. The third and fourth rows show *CriticAttack* results where the attacker uses a logistic regression classifier and a deep neural network, respectively.

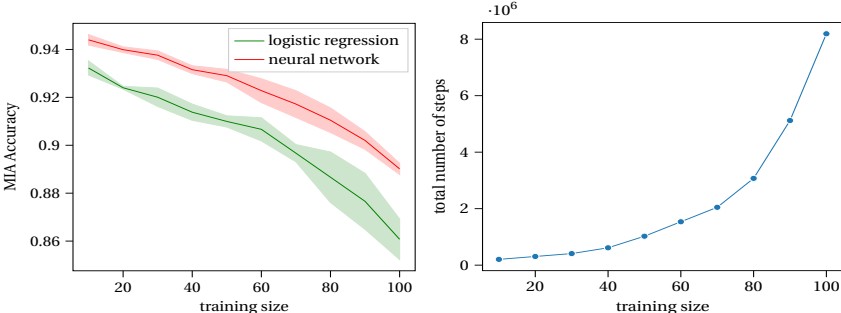

Figure 6: the left figure MIA results on target agents with varied numbers of training environments. These target agents are trained by GAE advantage for the Multi-Room task. The right figure shows the number of steps required for convergence.

**Definition A.1.** Let $f$ be a given function. Define two datasets $D$ and $D'$, both in the input domain of $f$, *adjacent* if the number of entries in which the two datasets hold different values is at most one.

After defining the adjacent dataset, we can define $(\epsilon, \delta)$-differential privacy as the following:

**Definition A.2** (($\epsilon, \delta$)-DIFFERENTIAL PRIVACY)**.** . Let $\epsilon$ be a positive real number and $M$ be a randomized algorithm that takes a dataset as input. Let $\mathcal{Y}$ be the image of $M$. The algorithm $M$ is $\epsilon$-differentially private if, for all adjacent datasets $D_1$ and $D_2$, and all $R \subseteq \mathcal{Y}$:

$$\mathbb{P}[M(D_1) \in R] \leq \exp(\epsilon)\mathbb{P}[M(D_2) \in R] + \delta, \tag{11}$$

where $\delta$ captures the probability that $\epsilon$-differential privacy fails. If $\delta = 0$, then we say $M$ is an $\epsilon$-**DIFFERENTIAL PRIVACY**.

**DP-LSL**   (Balle et al., 2016a; Lebensold et al., 2019a) is a pretraining protection mechanism that construct a differentially private value function (critic network). It achieves differential privacy by adding Gaussian noise to the critic's parameters before running the actor-critic algorithm.

In the initialization phase, we construct a differentially private critic in the actor-critic algorithm by adding Gaussian noise to the critic's parameters. DP-LSL refers to the process of adding noise, which takes the differential privacy budget $\epsilon$ and $\delta$ as input.

Then, we apply *CriticAttack* to the actor-critic algorithm with the differentially private critic. We present the MIA results in Table 3 and show that the DP-LSL has a negligible effect on protecting against *CriticAttack*. Note that we obtain the results by performing MIAs on the actor-critic algorithm with GAE.

The work (Lebensold et al., 2019a) empirically shows that DP-LSL can achieve differential privacy with minimal loss in performance. However, our experiments demonstrate that DP-LSL does not affect protecting against *CriticAttack*. We present the results on Table 3.

| $\epsilon$ | $\delta$ | LR MIA | DNN MIA | Final Reward |
|---|---|---|---|---|
| 10 | 2e-5 | $90.2 \pm 1.1$ | $91.1 \pm 1.3$ | $0.973 \pm 0.012$ |
| 1 | 2e-5 | $90.5 \pm 1.6$ | $90.6 \pm 1.8$ | $0.967 \pm 0.009$ |
| 0.1 | 2e-5 | $89.8 \pm 1.4$ | $91.3 \pm 1.5$ | $0.955 \pm 0.014$ |

Table 3: MIA accuracy and performance loss under DP-LSL. We report the (mean ± standard deviation) tuple across five repetitions.

**DP-SGD**   (Abadi et al., 2016) is a standard approach that helps deep learning models satisfy differential privacy. DP-SGD modifies the stochastic gradient descent in the training algorithm to enforce differential privacy to the algorithm itself. DP-SGD modifies the stochastic gradient descent in the training algorithm of the deep learning model to enforce differential privacy to the algorithm itself.

During the training procedure, DP-SGD first clips the gradients computed over the training data; then applies the Gaussian mechanism to add statistical noise drawn from a defined Gaussian distribution to the gradients; finally updates the model with the noisy gradients. Let $\theta$ be the parameters of the deep learning model; the DP-SGD works as the following:

$$\theta_{i+1} = \theta_i - \frac{\alpha}{\beta} M_{gauss}\left(\sum_{j=1}^{\beta} g(\nabla_\theta \mathcal{L}_j(\theta), C), \sigma\right), \tag{12}$$

where $\theta_i$ is the parameters of the deep learning model at iteration $i$; $\alpha, \beta$ are the learning rate and the batch size of the training algorithm; $g(x, C)$ is the clipping function defined by

$$g(x, C) = x \cdot \min\left(1, \frac{C}{\|x\|}\right).$$

$C, \sigma$ are the clipping value and the noise standard deviation of the DP-SGD algorithm.

We define the Gaussian mechanism as $M_{gauss}(f(x), \sigma) = f(x) + n$, where $n \sim \mathcal{N}(0, \sigma^2 \mathbf{I})$.

| Estimation(Attacker) | Multi-Rooms | Door-Key | Lava-Crossing | Four-Rooms |
|---|---|---|---|---|
| TD Advantage (LR) | 55.6(8.9) | 52.1(6.7) | 53.8(11.2) | 56.3(7.5) |
| TD Advantage (DNN) | 62.4(8.9) | 64.6(6.7) | 58.1(11.2) | 65.2(7.5) |
| N-Step (LR) | 56.3(9.9) | 50.8(7.5) | 58.1(11.9) | 54.2(8.8) |
| N-Step (DNN) | 63.6(9.9) | 59.7(7.5) | 64.1(11.9) | 58.4(8.8) |
| GAE (LR) | 53.7(8.4) | 51.5(7.2) | 55.9(9.7) | 52.6(8.1) |
| GAE (DNN) | 64.4(8.4) | 60.8(7.2) | 65.7(9.7) | 62.5(8.1) |

Table 4: MIA accuracies (corresponding performance loss in percentage) across five repetitions.

**DP-SGD Privacy Budget** We apply the DP-SGD algorithm to enforce differential privacy to actor-critic reinforcement learning. We set the gradient clipping value $\epsilon_{clip} = 0.2$, $epoch = 4$, batch size $\beta = 256$, privacy offset $\delta = 1e - 4$, and sample size $n = 1000$. We show the privacy budget $\epsilon$ at each noise variance $\sigma$ in Figure 7.

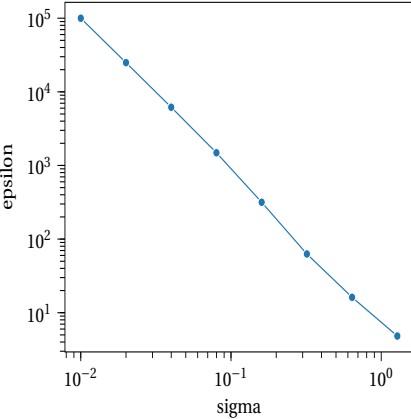

Figure 7: DP-SGD privacy budget at each noise level.

### A.7 EXAMPLE OF DEFENSE AN ENVIRONMENT

We apply run several agents on the same environment $E_0$ to collect value and reward-to-go trajectories. The first agent is unprotected and has experienced $E_0$ in its training phase, which means $E_0$ is 'in' the training dataset of the first agent. The second, third, and fourth agents are agents protected by DP-SGD with different noise variances during training. $E_0$ is 'in' the training dataset of these three agents. The fifth agent is unprotected and has not experienced $E_0$ during training. We present five sets of trajectories in Figure 8, respectively.

The correlations between the value and reward-to-go trajectories in the five rows are 0.88, 0.83, 0.71, 0.32, and 0.26.

We observe that a smaller amount of noise does not affect the correlation between the rewards-to-go and values, as shown in the second and third rows in Figure 8. Instead, a small amount of noise only changes the trajectories' smoothness. We must increase the noise variance to reduce the correlation, as in the third and fourth rows in Figure 8. However, we already significantly degrade the agent's performance by introducing a large noise.

### A.8 MORE DEFENSE RESULTS

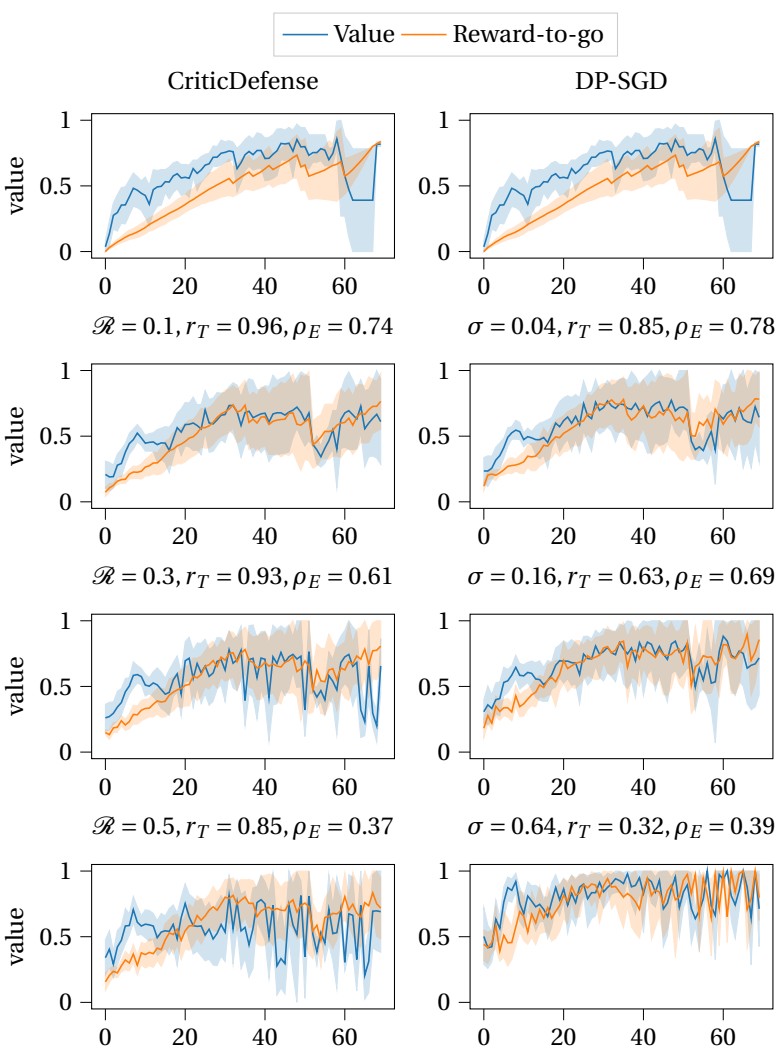

Figure 8: Each figure shows the value and reward-to-go trajectories from a selected environment under various conditions.

