# OpenReview forum: "Value-Based Membership Inference Attack on Actor-Critic Reinforcement Learning"
_ICLR.cc/2023/Conference — Submitted to ICLR 2023_

### Official Review · Reviewer_Xm55 · 2022-10-24

**Confidence:** 3
**Correctness:** 2
**Technical Novelty And Significance:** 3
**Empirical Novelty And Significance:** 1
**Recommendation:** 5

**Clarity, Quality, Novelty And Reproducibility:**

Quality: fair (refer to the weaknesses above)
Clarity: fair (the presentation of CriticAttack can be improved by using both description and pseudocode)
Originality: good

**Strength And Weaknesses:**

Strength:
1. This paper relaxes the assumptions in existing work by providing a black-box attack;
2. The idea of examing the correlation between expected rewards and value function is interesting;
3. Both attack and defense are studied.

Weakness:
1. there are not enough explanations for some steps in the proposed methods:
a. In CriticAttack: why performance on validation environments is used as stopping criteria?
b. In CriticAttack: when does the correlation technique work? Do we need to assume the target agent is trained with a set of similar environments and the other environments are quite different?
c. In CriticDefense: why do we use uniform noise instead of zero-mean noise (e.g. Gaussian)?
d. In CriticDefense: why do we use modulo instead of clipping to make sure the values are between 0 and 1? I understand modulo allows a more significant change, but I also think modulo could result in a worse policy.

2. There is no comparison in terms of performance between CriticAttack and existing methods. More experiments are needed to understand whether the 90% accuracy is a good accuracy.

**Summary Of The Paper:**

This paper studies the vulnerability of actor-critic reinforcement learning in the context of membership inference attacks. The attacker aims to make inferences about the training environments based on the outcomes of an RL algorithm. From the attack perspective, this paper proposes CriticAttack, which examines the correlation between the expected reward and the value function. CriticAttack only requires knowledge about the value function and expected rewards, which requires less information than prior work. From the defense perspective, this paper proposes CriticAttack, which inserts uniform noise to the value function to reduce correlation. Both CriticAttack and CriticDefense are evaluated empirically.

**Summary Of The Review:**

This is an interesting paper, but some steps in the main methods are not well-supported.

---

> ### Author Response · Authors · 2022-11-16
> **Response to Reviewer Xm55**
>
> We thank the reviewer for the insightful comments and questions. We conducted additional experiments and added the results to the paper; the details are explained below.
>
> 1a. As I mentioned at the beginning of section 4.2, we create a shadow agent to mimic the target agent. “Mimic” means that the shadow agent should perform as good as the target agent in new environments they have never seen before, which are the validation environments.
>
> 1b. The correlation technique works when the RL agent is overfitted to its training environment (which is quite common). For each task, we assume the training and validation environments are sampled from the same data source; therefore, all the environments are similar.
>
> 1c. Gaussian noise is ineffective in breaking the correlations unless we choose a very large variance.
>
> 1d. It is true that modulo allows a more significant change but results in a worse policy. That is the privacy-performance trade-off I mentioned in Section 6.2. Modulo provides better privacy protection but sacrifices performance. We use modulo because we care more about privacy.
>
> 2. We include some comparisons in Appendix A.4.

---

### Official Review · Reviewer_x7wn · 2022-10-25

**Confidence:** 2
**Correctness:** 3
**Technical Novelty And Significance:** 3
**Empirical Novelty And Significance:** 3
**Recommendation:** 6

**Clarity, Quality, Novelty And Reproducibility:**

As mentioned by the authors, CriticAttack collects values from the value function and the cumulative reward for membership inference, and Gomrokchi et al. (2021) and Gomrokchi et al. (2020) introduce two membership inference attack methods to infer the roll-out trajectories in off-policy RL algorithms, which learn the optimal policy independently of the agent’s actions. I am not sure how hard to transfer from a trajectory-based membership inference attack to an environment-based based membership inference attack proposed in this paper.

**Details Of Ethics Concerns:**

If the proposed scenario is realistic, the RL models may be attacked.

**Strength And Weaknesses:**

This paper proposes a new attack with a simple defense method to mitigate such an attack, which is evaluated by experiments. The main evidence of this paper is experimental evaluation, but all experiments are done on a single task (i.e., reach a target destination without bumping into obstacles) on MiniGrid toolkit. Let alone the proposed attack only works for actor-critic algorithms, and it is questionable if the proposed attack can attack other tasks.

I am not sure if the proposed attack is realistic. If the training datasets are known, why is it interesting to infer whether a data record is used in training the shadow model?
If the training datasets are known, we can train the target model. Why do we need to infer the target model from the trained attacker?

About environment-based MIA and trajectory-based MIA, which
one is easier? This paper focuses on environment-based MiA, but I think trajectory-based MIA is harder because an environment could have many trajectories. In addition, why do we care about inferring the environment when the information about the environment of the model is usually public?



**Summary Of The Paper:**

This paper studies how to launch the value-based membership inference attack on actor-critic reinforcement learning. Such attacks may make inferences about the training environments—whether a particular environment has been used in training—by observing the outcomes of a reinforcement learning algorithm.
They develop CriticAttack, a new membership inference attack that targets black-box RL agents by examining the correlation between the expected reward and the value function. They empirically show that CriticAttack can correctly infer approximately 90% of the training data membership by using the MiniGrid toolkit. To defend against CriticAttack, they designed a method called CriticDefense that inserts uniform noise into the value function. CriticDefense can reduce the attack accuracy to 60% while degrading no more than 10% of the agent’s performance.

**Summary Of The Review:**

This paper proposes a new attack with a simple defense method to mitigate such an attack, which suffers some limitation on the novelty, the significance of the work, and experimental evaluation.

---

> ### Author Response · Authors · 2022-11-16
> **Response to Reviewer x7wn**
>
> We thank the reviewer for the insightful comments and questions. To address the questions and concerns raised by the reviewer, we add the following explanations:
>
> We perform experiments on four tasks; three are trying to reach a target destination without bumping into obstacles, and the other is trying to find a key to open a door. In the introduction, we referenced some works claiming MIA on RL could infer users’ vehicle routes or room layouts, and the tasks we chose can simulate room or warehouse layouts. There are more tasks we can try, but the other tasks may not have enough privacy implications (e.g., I don’t think there is a privacy concern on AlphaGo).
>
> There is a difference between the data source and dataset, the data source is the data distribution, and the dataset is a set of records sampled from the data source. We know the data source and can sample records from the source, but we do not know whether this data record is used in training the target RL agent. We assume the data source of the target agent is known, but the dataset is unknown.
>
> Yes, trajectory-based MIA might be harder. We have trajectory-based MIA accuracies, an average is 87%, in contrast to 92% environment-based MIA. We use environment-based MIA to simulate the case that infers room layouts. As I mentioned, the “environment is public” means the “data source” is known, not individual data records.
>
> The main difference between our work and the two works from Gomrokchi is not trajectory-based vs. environment-based. The main difference (contribution) is that we focus on the value function rather than the policy.

---

### Official Review · Reviewer_JgPB · 2022-10-29

**Confidence:** 4
**Clarity, Quality, Novelty And Reproducibility:** See the last part.
**Correctness:** 1
**Technical Novelty And Significance:** 1
**Empirical Novelty And Significance:** 2
**Recommendation:** 3

**Strength And Weaknesses:**

Pros:

1. Investigations on RL algorithms' vulnerability and corresponding attack and defense methods seem relevant.

Cons:

1. It is not very realistic to assume the release of the value function. The manuscript provided a reason that releasing the value function could be for the customer's fine-tuning, which is kind of a stretch. In practice it is the converse: The critic could be discarded once one finishes the training. The actor is the only component that is released.

2. A binary prediction of whether a set of trajectories are used in the training does not sound to be anywhere an attack. Of course some trajectories could carry sensitive information, but most of the trajectories wouldn't. The manuscript does not provide a characterization on how much sensitive information each trajectory would carry and how the prediction works in terms of sensitive information on could obtain. The so-called 90% accuracy also doesn't make much sense under this view.

3. Based on 2, the so-called defense also doesn't make sense, because this defense only protects one from this specific attack (which is not necessarily an attack). To be specific, adding some noise towards the value function only prevents one from figuring things out from the value function, but one could still possibly figure things out from the induced policy. The relevance of this defense is questionable.

4. The presentation of this manuscript is weird: Several proclaimed names are given without introducing the specific methods; The attack algorithm is introduced even before introducing the attack's goal; The algorithm is more like a list of existing policy evaluation methods; The manuscript is filled up by equations but I don't see them informative or even needed; Notations like % and $\oplus$ are used without definition; Weird usage of mathcal; etc.

**Summary Of The Paper:**

This manuscript proposes a binary classifier that determines if an environment is used in the training of an agent, by inspecting $n$ trajectories that are generated from that environment. The manuscript claims this to be an attack toward a value function of an RL agent. The manuscript subsequently proposes to perturb the value function so as to mitigate the capability of making the aforementioned binary classification, and proclaims this to be a defense method. Both the classifier and the prevention of the classifier are evaluated in an empirical manner.

**Summary Of The Review:**

The manuscript has spent a decent effort on trying to figure out an attack method towards RL but I do not see either the setting or the method making sense.

---

> ### Author Response · Authors · 2022-11-17
> **Response to Reviewer JgPB**
>
> We thank the reviewer for the insightful comments and questions. In order to address the questions and concerns raised by the reviewer, we conducted additional experiments and made some modifications to the paper.
>
> 1. We added several other reasons on why we think releasing value functions could be realistic in the last paragraph of the introduction. Teacher agents in transfer learning and teacher-student framework often need to release their value function.
>
> 2. A membership inference attack (MIA) is a well-known attack that does binary classification. And there are many works about MIA, even in reinforcement learning, which I mentioned in the related work. MIA is one of the most well-known and important privacy attacks in machine learning; I do not understand why the reviewer thinks that MIA is not necessarily an attack.
> As we mentioned in the first paragraph of the introduction, the attack may infer personal room layouts or vehicle paths, which are sensitive information. We added a comparison between our attack and some other existing attacks to show that 90\% accuracy is a high accuracy.
>
> 3. I have specified in Section 5.2 that the defense is specifically designed to protect the value function. And we do not consider the defense as one of our main contributions and do not aim to provide protection to all the components. The defense method is only a method associated with the attack, while the attack is the main contribution.
>
> 4. a) “Several proclaimed names are given without introducing the specific methods:” we either introduced in the Preliminary or cited the corresponding paper.
> b) “The attack algorithm is introduced even before introducing the attack's goal:” the name Membership Inference Attack already introduced the attack’s goal; we also mentioned it in the Preliminary.
> c) “The manuscript is filled up by equations but I don't see them informative or even needed:” I have removed some equations that may be considered common knowledge.
> d) I have gone through the notations and ensured every notation is defined.

---

> > ### Comment · Reviewer_JgPB · 2022-12-01
> > **Thank you for your response**
> >
> > I've read your response and I thank the authors for their effort in fixing some issues.

---

### Decision · Program_Chairs · 2023-01-20

**Decision:**

Reject

**Justification For Why Not Higher Score:**

NA

**Justification For Why Not Lower Score:**

NA

**Metareview: Summary, Strengths And Weaknesses:**

This paper studies the problem of Membership Inference Attacks (MIAs) on actor-critic RL methods. The paper introduces an empirical attack algorithm that can infer the membership of a given trajectory with good accuracy, given that the value function is released to the attack. As is pointed out by reviewer JgPB, empirically, it doesn't really make sense for the system designer to release such information in the first place. Overall, this paper seems to be half-baked and not ready for publication.